# Cell-Penetrating Antimicrobial Peptides with Anti-Infective Activity against Intracellular Pathogens

**DOI:** 10.3390/antibiotics11121772

**Published:** 2022-12-08

**Authors:** Gabriela Silva Cruz, Ariane Teixeira dos Santos, Erika Helena Salles de Brito, Gandhi Rádis-Baptista

**Affiliations:** 1Postgraduate Program in Pharmaceutical Sciences, Faculty of Pharmacy, Dentistry, and Nursing, Federal University of Ceara, Fortaleza 60416-030, Brazil; 2Laboratory of Biochemistry and Biotechnology, Institute for Marine Sciences, Federal University of Ceara, Fortaleza 60165-081, Brazil; 3Microbiology Laboratory, Institute of Health Sciences, University of International Integration of the Afro-Brazilian Lusophony, Redenção 62790-970, Brazil

**Keywords:** antimicrobial peptide, cell-penetrating peptide, cell-penetrating antimicrobial peptide, intracellular pathogen, intracellular infection

## Abstract

Cell-penetrating peptides (CPPs) are natural or engineered peptide sequences with the intrinsic ability to internalize into a diversity of cell types and simultaneously transport hydrophilic molecules and nanomaterials, of which the cellular uptake is often limited. In addition to this primordial activity of cell penetration without membrane disruption, multivalent antimicrobial activity accompanies some CPPs. Antimicrobial peptides (AMPs) with cell-penetrability exert their effect intracellularly, and they are of great interest. CPPs with antimicrobial activity (CPAPs) comprise a particular class of bioactive peptides that arise as promising agents against difficult-to-treat intracellular infections. This short review aims to present the antibacterial, antiparasitic, and antiviral effects of various cell-penetrating antimicrobial peptides currently documented. Examples include the antimicrobial effects of different CPAPs against bacteria that can propagate intracellularly, like *Staphylococcus* sp., *Streptococcus* sp., *Chlamydia trachomati*s, *Escherichia coli*, *Mycobacterium* sp., *Listeria* sp., *Salmonella* sp. among others. CPAPs with antiviral effects that interfere with the intracellular replication of HIV, hepatitis B, HPV, and herpes virus. Additionally, CPAPs with activity against protozoa of the genera *Leishmania*, *Trypanosoma*, and *Plasmodium*, the etiological agents of Leishmaniasis, Chagas’ Disease, and Malaria, respectively. The information provided in this review emphasizes the potential of multivalent CPAPs, with anti-infective properties for application against various intracellular infections. So far, CPAPs bear a promise of druggability for the translational medical use of CPPs alone or in combination with chemotherapeutics. Moreover, CPAPs could be an exciting alternative for pharmaceutical design and treating intracellular infectious diseases.

## 1. Introduction

Intracellular pathogens can grow and reproduce within their host cells, and this feature helps them evade the immune system and makes treatment difficult. Accessible and efficacious drugs targeting these pathogens are challenging, as many antimicrobials cannot accumulate in intracellular spaces to reach optimal therapeutic concentrations in infected cells [1]. Treatment options generally involve long-term therapy with a combination of medications. However, the drug’s inherent toxicity and prolonged exposure time of treatment usually cause undesired side effects. Therefore, strategies targeting intracellular infection and eradicating intracellular pathogens are advantageous since therapeutic agents penetrate and concentrate into infected cells, reaching the pathogens, increasing efficacy, decreasing drug toxicity, and sustaining and releasing the drugs [2]. Faced with these challenges, cell-penetrating peptides (CPPs) have emerged as an exciting alternative for pharmaceutical drug design and formulations. CPPs are known for their great potential for delivering peptides and proteins into cells and across epithelial barriers, both in vitro and in vivo [3]. Having the unique ability to transport various payloads within cells with low toxicity, CPPs hold the promise of a powerful tool for medical applications. They can transport molecules for which intracellular supply is often limited due to hydrophilic character, net negative charge, and high molecular weight. In addition, they can mediate the delivery of drugs, diagnostic agents, nanoparticles, and therapeutic proteins [4].

Notably, CPPs share several physicochemical characteristics with another class of biologically active peptides, antimicrobial peptides (AMPs). CPPs with antimicrobial activities and reciprocally AMPs with cell-penetrating activities are generally short sequences of amino acids, amphipathic, and display a net positive charge due to a high proportion of arginine and lysine residues. The sources of CPPs with antimicrobial activity are diverse: they might be natural, synthetic, or chimeric. The spectrum of action of cell-penetrating peptides with anti-infective properties includes activity against intracellular Gram-positive and Gram-negative bacteria, parasites, viruses, and fungi, as reviewed and discussed herein. Considering this initial conception, CPPs with anti-infective activities arise as an exciting alternative for pharmaceutical drug design to treat intracellular infections. This short review aims to call attention to and identify in the scientific literature previously published studies and references regarding the action of CPPs that are effective against intracellular infectious pathogenic agents.

## 2. Intracellular Pathogens

### 2.1. Bacteria

Many microorganisms are obligate or facultative intracellular, which allows growth and replication in different biological niches. When initiating the process of internalization in the host cells, some microbial pathogens can be detected by the host defense mechanisms, activating the recruitment of macrophages. However, as is the case, intracellular bacteria can protect themselves from the host’s immune response and the killing effects of antimicrobials inside the infected cells. In this way, intracellular bacteria reside in the cytoplasm or vacuoles of the mammalian host cell, coopting the endocytic or secretory pathways to recruit the necessary supplements to ensure their replication [5]. Pathogens can enter different cell types via phagocytic and non-phagocytic routes. Phagocytosis is a mechanism related to the host’s response to injuries and infections. Pathogens or scavengers larger than 0.5 μm are recognized by phagocytic cells through specific pathogen recognition receptors. Then, phagosomes that contain the microorganisms and scavengers form, usually in macrophages, that fuse with lysosomes to form mature phagolysosomes, where the bacterium is degraded due to lysozymes and oxidizing agents. *Mycobacterium tuberculosis* is one of the most successful pathogens in escaping this process. It has evolved to prevent phagolysosome maturation and can live in macrophages, masking pathogen-associated molecular patterns (PAMPs) [6]. A latent state may occur for bacterial and viral infections, such as those caused by *M. tuberculosis* and HIV.

Blocking the progression of the disease may not only result from the annihilation of pathogens but may also involve the death of the host cell. The disease process may involve interrupting bacterial growth in a subset of cells by innate immune pathways. However, it may trigger an inflammatory response that does not interfere with pathogen replication, as is seen with the enteric bacteria *Salmonella* during intestinal growth [7]. Another example of an intracellular microorganism, *Listeria* spp. can induce epigenetic and miRNA modifications in the host to modulate immune defense. *Listeria monocytogenes* is a Gram-positive facultative pathogen that causes the foodborne illness called Listeriosis. Upon entry into epithelial cells, *L. monocytogenes* are internalized into a vacuole. It physically disrupts the vacuolar membrane to escape this subcellular niche, using Listeriolysin O (LLO) and phospholipase A and B (PlcA and PlcB). Then, *L. monocytogeneses* can survive and replicate in the cytosol of host cells and modify the processes of cellular organelles. Intracellular bacteria can also grow an actin tail to move between epithelial cells evading host defense [8].

Chlamydiae, an obligate intracellular Gram-negative bacterium, has a unique developmental cycle of replication consisting of extracellular and intracellular life forms. *Chlamydia trachomatis* and *C. pneumonia* cause sexually transmitted diseases, eye infections, and atypical pneumonia. *Chlamydiae* have developed strategies for chromatin regulation through epigenetic modifications [9]. In addition to the classic examples of intracellular bacteria such as the case of *Listeria* and *Chlamydia*, cumulative evidence suggests that known extracellular bacteria such as *Staphylococcus aureus*, *Escherichia coli*, and *Pseudomonas aeruginosa* also can invade and localize inside the host cells [10].

### 2.2. Parasites

Regarding parasitic infections, intracellular obligate parasites use diverse strategies to deceive host cell immune responses and persist in the host cells and organism. One common strategy in clinically relevant infectious diseases is the formation of vacuoles containing pathogens within host cells after the pathogen has been internalized. Included in this category are the trypanosomatids, of which *Trypanosoma cruzi* and *Leishmania* spp. are intracellular parasites of mammals [11]. In the case of the causative agent of Leishmaniasis, the first hours after infection are crucial to the obligated intracellular *Leishmania* parasites. *Leishmania* parasites benefit from the pro-inflammatory properties of the mosquito vector’s saliva, which also plays an essential role in the chemoattraction of phagocytes.

Additionally, the complement system affects the initial phases of *Leishmania* infection, as opsonization promotes the parasite uptake by phagocytes, favoring the intracellular stage of infection. Once the parasite is phagocytosed, innate immune cells react by producing cytokines that, in its turn, activate the adaptive immunity, thus generating a protective or harmful response. Adaptive immune system cells have also participated in Leishmaniasis’s pathogenesis, causing tissue destruction and disease relapse [12]. For *Plasmodium falciparum*, one of the etiological agents of malaria, free sporozoites and intrahepatic parasites overcome the obstacle of the host’s immune response to entering the erythrocytic stage, actively passing through Kupffer cells and endothelial cells. Once inside the hepatocyte, the parasitophorous vacuole prevents lysosomal degradation. Host heme oxygenase-1 (HO-1) also enhances the development of intrahepatic parasites by modulating the host’s inflammatory response [13].

### 2.3. Viruses

As for viruses, such as those caused by Human Immunodeficiency Virus (HIV) from type HIV-1 and HIV-2, Hepatitis B Virus, Influenza Virus, to Middle East Respiratory Syndrome Coronavirus, SARS-CoV-2 and others, the high number of cases of these viral diseases are related to several environmental and behavioral factors, which contribute to the spreading of these highly persistent and replicative viruses. Mutations constantly appear in the genomes of all species, making it difficult to treat and control some viral diseases [14].

## 3. Challenge to Treat Intracellular Infections

Efficient treatment of intracellular infections with antimicrobials is challenging due to the evasion of intracellular infectious pathogens from the host’s phagocytic killing mechanisms, the establishment of intracellular survival machinery, and the misuse of antimicrobials that cause an increase of multidrug resistance phenotypes in pathogens. Once inside the cell, antimicrobial activity can be influenced by enzymatic inactivation and changes in pH and chemical environment [15]. Moreover, the intracellular concentration of antibiotics in host cells is lower than their minimum inhibitory and lethal concentrations, which may lead to the emergence of drug resistance. The increase and spread of multidrug-resistant bacteria and infectious agents reinforce the urgent need to develop effective intracellularly active antibacterial drugs and chemotherapeutics capable of improving the low cellular permeability of currently used antimicrobials.

In this context, CPPs with anti-infective activities have been envisioned as an alternative to overcome the difficulty-to-treat intracellular infectious microbes and to potentiate the targeted-driven antimicrobial action to defeat intracellular microorganisms [16,17].

## 4. Cell-Penetrating and Antimicrobial Peptides: The Basics

Cell-penetrating peptides are natural or synthetic peptide sequences, usually displaying selective cytotoxicity, high cell-uptake efficiency, and penetrability of lipid membranes and accumulation in intracellular compartments. Another notable feature of CPPs is that cell membrane translocation can occur at low concentrations, both in vivo and in vitro, without causing significant membrane damage. Cell-penetrating peptides usually consist of 5 to 30 amino acid residues. They group into different categories according to their origin, sequence, and structure [18]. One of the first CPPs ever described was the HIV transcriptional transactivator protein (TAT) which was proven to translocate cell membranes and effectively internalize into the cells in vitro. The TAT peptide becomes a paradigmatic CPP for basic and applied research [19,20]. Endocytosis and direct translocation are essential mechanisms contributing to membrane translocation by different CPPs. These events are related to the specific CPP sequence, peptide concentration, cell type, and cell differentiation state, among other factors [21]. Several CPPs adopt a well-defined secondary structure upon encountering the cell plasma membrane, which seems to contribute to translocation propensity across the membrane [22]. Some CPPs and AMPs share structural and physicochemical characteristics that intriguingly call attention to their subtle differences and eventual similar functionalities (e.g., cell penetration and anti-infectivity) [23,24]. Native AMPs are widely found in organisms’ cells, tissues, and biological fluids and are essentially involved in the innate defense of the host against invading pathogens. They kill microbes by diverse but convergent mechanisms of cell disruption that essentially compromise the integrity of the cytoplasmic membranes [25,26], which serve as a portal for the synergic action of chemotherapeutics [27]. A handful of exciting reviews about the structures, activities, and mechanisms of action of CPPs and AMPs [28,29,30] are available. Some suggested reviews on the biological activity of AMPs against cancer cells [31,32,33] and CPPs as delivery systems [34,35,36,37] could be of interest.

The cellular penetrability of some AMPs seems to reinforce their antimicrobial efficacy and performance due to the interaction and interference with intracellular components of target microorganisms, such as macromolecules and organelles [31,32]. Consequently, AMPs interacting with cell membranes can exhibit antimicrobial activity at the membrane level or intracellularly through cell-penetrability. The positively-charged amino acid residues that facilitate the electrostatic interactions of CPPs and AMPs with cell membranes and their components are common to sequences of both peptide types. Interspaced positively charged amino acid residues in these two peptide families contribute to the first step of membrane interaction and aggregation, leading to membrane insertion and cellular uptake. Components on the cell surface with an overall net negative charge facilitate binding the positively charged and amphipathic peptides. These are the anionic phospholipids or phosphate groups of lipopolysaccharides in Gram-negative bacteria and acidic polysaccharides, teichoic acids, and lipoteichoic acids in Gram-positive bacteria [33].

The antiviral mechanisms of AMPs mainly involve targeting and disrupting viral membrane envelopes and inhibiting different stages of the viral life cycle [33,34]. As for the majority of therapeutic peptides, the physiological instability, lack of selectivity, and limited efficacy comprise drawbacks that should be considered during the development and medicinal use of CPPs with antimicrobial activity and AMPs with cell-penetrating properties. Proteolytic degradation, hepatic and renal clearance, and inherent instability of CPPs and AMPs in physiological fluids are reasons for the large discrepancy of their performance in vitro and in vivo [35,36].

## 5. Peptides with Cell-Penetrating and Anti-Infective Activity against Intracellular Pathogens

In the current scientific literature, cumulative data about cell-penetrating antimicrobial peptides (CPAPs) with anti-infective activities to treat infections caused by obligatory or facultative intracellular pathogens are accessible. This particular class of bioactive peptides is also called antimicrobial and cell-penetrating peptides (ACPPs) [37]. Antimicrobial peptides that penetrate the target cells without causing membrane disruption exert their biocide effects intracellularly figure in the subset of CPAPs [38,39]. CPPs that intrinsically translocate eukaryotic cell membranes and cross the cytoplasmic membrane of bacteria can kill pathogens by distributing and accumulating within the vacuolar compartments of cells, as is the case of penetratin analogs [40]. The design and synthesis of chimeric CPP-AMP peptides with a cleavable link also comprise an attractive and efficient mechanistic option to kill intracellular pathogens [41,42]. Another exciting mechanism to kill intracellular pathogens is via a cooperative action of the AMPS α-defensins and humanized θ-defensin with macrophages that entrapped the bacteria in the phagosome, where they are annihilate [43]. The following sections present examples of these cell-penetrating antimicrobial peptides with antibacterial, antiviral, and antiparasitic activities.

### 5.1. Cell-Penetrating Antimicrobial Peptides Active against Bacteria

Essentially, bacteria evolved diverse strategies to escape from the antimicrobial arsenal of macrophages, and once bacteria internalized into macrophages, they evade the host’s immune response and the action of antimicrobial drugs [44]. The first barrier for a therapeutic compound to reach intracellular bacteria is crossing the host cell membrane at non-toxic concentrations. The second barrier is the challenge of accessing bacteria located in the host cell cytosol or sequestered inside membrane-bound vesicles and maintaining an effective antimicrobial activity. Several CPAPs have been investigated for their applicability against intracellular bacteria in this context.

The Gram-positive bacterium *S. aureus* is considered an extracellular pathogen. However, it can survive in primary human macrophages until bacterial proliferation occurs. Replication begins hours after the initial phagocytosis, while the bacteria reside in mature phagolysosomes and may remain for several days. The lysis of infected phagocytes allows them to escape to the extracellular medium, where dissemination occurs [45]. Budagavi et al. [46] evaluated the antimicrobial properties of cell-penetrating peptides derived from Latarcin1 (LDP) toxin against methicillin-resistant *S. aureus* (MRSA) strains. Latarcins are a group of antimicrobials and cytolytic peptides, 20 to 35 amino acids in length, from the venom of the *Lachesana tarabaevi* spider. Human cervical cancer epithelial (HeLa) cells were used for the invasion assay by MRSA strains and *Bacillus subtili*s, *E. coli*, *S. typhimurium*, *M. smegmatis,* and *Xanthomonas oryzae*. The peptide exhibited antimicrobial activity against all microorganisms tested, within a range of 5 to 20 μM, showing a 75% inhibition of intracellular MRSA at different concentrations. However, the peptide’s mechanism of action was not fully described [46]. Other studies have also attempted to identify peptides with activity against intracellular MRSA infections, such as the TAT peptide (47–58) (YGRKKRRQRRRD) derived from the TAT protein of the HIV-1 virus, responsible for activating viral replication. The authors studied this paradigmatic peptide because of its ability to penetrate the fungal cell membrane in a time- and temperature-independent manner, leading to cell cycle arrest and death. Infected HeLa cells were treated with TAT (47–58) and its enantiomer at a concentration of 50 μM and observed an antibacterial activity against MRSA without affecting the viability of mammalian host cells [47]. Wang and colleagues [48] investigated the antibacterial activity of the H2 peptide against MRSA strains causing bovine mastitis. They confirmed that the peptide enters into mammary epithelial cells by clathrin-mediated endocytosis in a dose-dependent manner, killing the intracellular MRSA and clinical isolates of *S. aureus* with a performance superior to vancomycin. Another Gram-positive coccus, *Streptococcus agalactiae*, can survive within macrophages for prolonged periods, contributing to pathogen spread and disease progression. To defeat *S. agalactiae*, peptides L2 and L10 were developed and exhibited more potent antimicrobial activity against bacteria inside the cell in an intracellular environment. The results of the localization of peptides in macrophages indicated that after endocytosis, L2 and L10 probably entered the compartments containing *S. agalactiae* by endosomal traffic and killing the bacteria [42]. Nepal et al. [49] attempted to demonstrate that increasing the length of CAPH peptides (i.e., CPAPs based on a cationic amphiphilic proline helix structure) increases their ability to enter cells and the antimicrobial activity compared to shorter cationic amphiphilic proline helix structures. This peptide has a dual mode of action: effective cell penetration capability into human macrophages and potent antimicrobial activity in vitro against Gram-positive and Gram-negative pathogens. Several pathogenic bacteria were effectively killed with the Fl-PRPLPL-5 peptide, including MRSA, *Acinetobacter baumannii*, *E. coli* O157, and VRSA (Vancomycin-resistant *S. aureus*). Significantly, this CPAP could deal with the problem of intracellular bacteria and the elimination of pathogens in macrophages [49]. Three similar peptides, also of the same class, derived from the initial peptide P14LRR, have specific subcellular locations that allow the targeting of pathogenic bacteria in their intracellular niches. This unique feature successfully eliminates *Salmonella*, *Shigella*, and *Listeria pathogens*, which reside in macrophages. More effective subcellular localization of P14-5C within endosomes may allow this CAPH to localize to endosome-resident *Salmonella*, thus resulting in improved intracellular clearance of the pathogen. The P14-5L peptide accumulates in the cytosol of macrophages and significantly reduces the population of *Listeria* and *Shigella* residing in the cytosol of J774A cells. The selected CAPHs could substantially reduce *Listeria* bacterial infection in an in vivo model of *Caenorhabditis elegans* with minimal toxicity to the worms [50]. The Fl-PLPRPR-4 peptide, another proline-rich synthetic peptide, displayed activity to kill intracellular bacterial pathogens, *Salmonella* and *Brucella*, infecting J774A cells. While intracellular *Salmonella* was reduced by approximately 62% with the addition of the peptide, intracellular *Brucella* showed a significant reduction of 90%. This difference in inhibition between *Salmonella* and *Brucella* may be related to several factors but the subcellular location of *Salmonella* and compartmentalization inside phagosomes. These synthetic peptides could internalize cells by endocytosis, direct translocation, or a combination of both pathways, depending on concentration, cell type, and charge. The entry of Fl-PLPRPR-4 in both endosomes and the mitochondria of J774A.1 cell may be related to both penetration mechanisms, by which peptide reach bacteria residing in infected J774A.1 cells, but with a limited impact on mitochondrial function [51]. Another example of peptide, TPk was investigated to verify its ability to internalize in eukaryotic cells, the intracellular location, the level of growth inhibition of bacterial growth, the influence of eukaryotic cell viability, and the destabilization of membranes through the use of liposomes that mimic *S. aureus* and mammalian cell membranes. TPk is transported to lysosomes and is considered a candidate for future investigations as an AMP capable of targeting intracellular infections, as it can potentially be optimized for cellular penetration, resulting in an effective tool for eradicating bacteria residing within epithelial cells [16].

*Chlamydia trachomatis* can survive and replicate in an intracellular or extracellular environment. The synthetic peptide Pep-1 contains a hydrophobic tryptophan-rich motif and a hydrophilic lysine-rich domain for efficiently targeting cell membranes. Pep-1 had a concentration-dependent action against intracellular *C. trachomatis* growth with 100% inhibition of inclusion formation at a concentration of 8 mg/L, with a window of susceptibility during the developmental cycle with maximum effect when treatment was initiated 12 h after infection. This anti-chlamydial activity may directly affect the bacterium or indirectly by targeting the inclusion of *C. trachomatis* or host cell processes essential for growth [52]. Amiss and colleagues [53] identified two horseshoe crab β-hairpin peptides, tachyplesin I and polyphemusin I, with broad antimicrobial activity against *E. coli* UPEC (Uropathogenic *E. coli*), strain EC958. They also developed the peptide analogs [I11A]tachyplesin I and [I11S]tachyplesin I that maintained activity against bacteria but were less toxic to mammalian cells than native tachyplesin I. For the in vitro tests, macrophages derived from the bone marrow of mice were used. Polyphemusin I, tachyplesin I, and analogs [I11A]tachyplesin I and [I11S]tachyplesin I inhibit bacterial growth at concentrations at least ten-fold lower than concentrations that are toxic to mammalian cells (8 µM). EC958 were observed within intracellular vesicle compartments, and after treatment of infected macrophages with eight µM of the marked peptides for one hour, especially [I11S]tachyplesin I produced diffuse fluorescence throughout the cytosol and strong co-localization with intracellular EC958 bacteria. This study demonstrated that peptides could enter macrophages without damaging their membrane at concentrations that kill co-localized bacteria. The mechanistic studies using bacterial cells, model membranes, and cell membrane extracts, indicated tachyplesin I and polyphemusin I peptides could kill UPEC by selectively binding and disrupting bacterial cell membranes.

Table 1 summarizes active CPAPs against intracellular bacteria, their amino acid sequences, origins, and targeted bacteria.

### 5.2. Cell-Penetrating Antimicrobial Peptides Active against Virus

Usually, the penetration of hydrophilic drugs that act intracellularly has their effect impaired due to the lipophilic barrier of the cell membranes, resulting in poor efficacy on their targets. Many techniques have been developed to find a non-invasive carrier to circumvent this problem and increase drug availability. Antiviral CPAPs can enter cells without causing appreciable damage, increasing molecules’ cellular uptake and internalization [54].

HIV has been the subject of numerous studies for a long time to seek a cure for carriers of this virus. Recently, a synthetic 18-residue cationic peptide capable of neutralizing the DH12 and SF162 strains of the HIV1 virus showed inhibition of 80% for both virus strains. In addition, its action was potentiated when using the bacterial toxins LT-IB or LT-IIaB derived from *E. coli*. This peptide was found to home in the cytoplasm and nucleus, demonstrating that it can act and influence the stages of production and maturation of the HIV-1 virus [55]. In another example, the 10-residue TAT peptide derived from the stretch comprising residues 48 to 57 of the TAT protein found in the HIV-1 virus can inhibit HIV-1 viral infection in a dose-dependent manner. 

Moreover, a change in the peptide net charge can increase its activity with the replacement of arginine by non-cationic residues, and the conjugation of deca-arginine potentiated the action of the TAT peptide [56]. RNA plays a role in regulating several processes in the biological system. In HIV-1, it is possible to find TAR, a short hairpin of RNA. This hairpin interacts with the TAT protein, responsible for the viral gene transcription, and this interaction has been the research subject in developing drugs that act at this point. The LK-3 peptide was synthesized with leucine (L) and lysine (K) residues, and results showed that in nanomolar concentrations, the LK dimer effectively inhibits TAR transcription, thus being a promising agent with strong HIV-1 inhibitory potential [57]. A determining agent for the assembly of the HIV-1 virus is the GAG protein, an envelope of this virus, thus being considered a critical pharmaceutical target. A peptide called CAI was discovered from bacteriophages and in vitro studies showed its ability to inhibit the formation of the GAG capsid. However, it was ineffective in HIV-1. After structural analysis, this peptide underwent modifications to make it cell penetrable, thus creating NYAD-1. This peptide demonstrated significant effects, such that it enters cells without the aid of a carrier protein, stops the development of HIV-1 particles, and effectively inhibits the HIV-1 infection in cell cultures [58]. Another molecule acting on HIV-1 was discovered after genomic analyses: the TOE1 protein, a target of the Egr1 gene, found in the nucleoli and Cajal’s bodies. In vitro studies, this protein has been shown to act at the transcriptional level, preventing replication of HIV-1 so that this protein can be released by CD8+ cells penetrating cells, thus decreasing the action of HIV [59].

The Human Papillomavirus (HPV) is responsible for causing an average of 5% of cervical cancer, and although vaccines are available, they are efficacious against only one set of HPV. Understanding the life cycle and performance of HPV can lead to the production of new pharmacological strategies that reduce its viral load and other HPV-related diseases. HPV is a DNA virus with 360 L1 capsid residues and 72 L2 molecules. The L2 protein is essential for the assembly and infectious viral process. A 29-residue peptide was synthesized, containing a binding site for L2 and a sequence for cell penetration into cells. Peptide P16/16 has been shown to inhibit the release of the virus from the endosome, causing a reduction of viral constituents in cells which is dose-dependent, thus being a potential antiviral target [60].

The herpes virus (*Herpes simplex* type—HSV-1) can trigger diseases in the human body ranging from cold sores to encephalitis and blindness. Two peptides (Hp1036 and Hp1239) derived from the scorpion *Heterometrus petersii* significantly inhibit the cell entry and proliferation of the HSV-1 virus. The amphipathic characteristic of peptides and their α-helix structure helped insert the virus into the membrane and reduce its replication after infection. Based on these data, the peptides Hp1036 and Hp1239 appeared promising candidates to combat viral activity [61]. Belonging to the Hepadnaviridae family with a size of 3.2 kb, the hepatitis B virus (HBV) is one of the leading infectious human viruses, affecting millions worldwide annually. HBV infection can cause chronic hepatitis and liver cancer. Therapy for hepatitis B is limiting, and most macromolecular antivirals cannot cross the impermeable plasma membrane. Based on the necessity to treat this viral disease, a peptide was synthesized that comprises an oligoarginine and a core capsid binding sequence (NBS). Such a peptide effectively penetrated the plasma membrane, inhibiting the release of the hepatitis B virus, proving to be an efficient transducing agent with intrinsic activity for delivering antiviral peptides to cells [62].

Table 2 lists some examples of cell-penetrating antimicrobial peptides with antiviral activity.

### 5.3. Cell-Penetrating Antimicrobial Peptides Active against Intracellular Parasites

The search for antimicrobial, cell-penetrating peptides with antiparasitic effects is of great interest since some infectious parasites hide inside host cells, and drug resistance increases considerably and concomitantly with drug ineffectiveness.

Among diseases caused by parasites, malaria is a potentially severe infectious disease transmitted by the bite of the Anopheles mosquito, which affects an average of one million persons annually. Of particular interest, antimalarial peptides have been discovered in recent years. Crotamine, a 42-amino acid, a multivalent peptide derived from the venom of South American rattlesnake *Crotalus durissus terrificus*, was investigated as an inhibitor of the *Plasmodium falciparum* parasite, and it was found that this substance prevents the parasite development in a dose-dependent manner. Its action may be related to the arrest of homeostasis in the acidic compartments of *Plasmodium*. In addition, the peptide decreased the fluorescence of dye tracker in the parasite’s organelle, indicating a compromise of the parasite’s metabolism [64].

Moreover, crotamine internalizes into erythrocytes infected by *P. falciparum* as confirmed with exposed cultured cells harboring blood-stage development of the parasite and the peptide in different incubation periods. Crotamine has no appreciable hemolytic activity and is selectively internalized only by infected erythrocytes [65]. Another peptide, derived from *P. falciparum* itself, demonstrated an antiparasitic effect. *Plasmodium falciparum* dihydrofolate reductase-thymidylate synthase (pfDHFR-TS) is a bifunctional enzyme responsible for folate production and thymidylate (dTMP). The pfDHFR-TS comprises 231 residues of the DHFR domain in the N-terminal region, followed by a junctional region (JR) with 89 residues and ending with the C-terminal end of 238 residues [66]. Peptide sequences from the junctional region and DHFR domain five peptides were designed and synthesized. Among these, the peptides JR21 and rR8-JR21 were effective in inhibiting the growth of *P. falciparum* in vitro. The rR8-JR21 peptide effectively delayed parasite development and caused minimal hemolysis of red blood cells, thus being a substance capable of affecting the development and growth of the parasite [67].

The transportan 10 peptide (TP10) is a 27-residue chimeric CPP derived from the venom peptide of the wasp *Vespula lewisii*. This peptide can penetrate the cell independently of the receptor and transport several types of cargo across cell membranes [68]. TP10 was tested for its anti-plasmodial and anti-trypanocidal potential. In vitro cell culture, this peptide reduced parasitemia by more than 99% and decreased oocyst infection, thus being an active substance against malaria in the blood and vector stages. Against the vector, *Trypanosoma brucei*, TP10 was also detrimental to the parasite in the blood phase, thus emerging as an anti-plasmodial and anti-trypanocidal peptide [69]. A developed mitochondria-penetratin peptide (MPP) with the ability to enter human cells and specifically target the mitochondria effectively controlled intracellular *Plasmodium* parasite. In some instances, mitochondria are considered an exciting drug target because they play a crucial role in energy production and programmed cell death of eukaryotic cells. The combination of two significant characteristics, cationic and lipophilic, drives the permeation of MPPs through the hydrophobic mitochondrial membrane. A short peptide, (L-cyclohexyl alanine-D-arginine)^3^ or (Fxr)^3^ exhibited efficient cell penetration and mitochondrial localization with activity against *P. falciparum*, including chloroquine-resistant K1 strain. This peptide can pass through the red blood cell membrane without disruption or destruction and subsequently kill the blood stage of *P. falciparum*. (Fxr)3 showed more potent antimalarial activity toward late-stage (trophozoite and schizont) parasites, consistent with the high intensity of (Fxr)3 localized in the parasites’ mitochondria observed by confocal microscopy. The antimalarial action may be related to a collapse of the mitochondrial membrane potential [70].

Trypanosomiasis, popularly known as Chagas’ disease, is caused by the protozoan *T. cruzi* and is among the 13 most neglected diseases worldwide, affecting about 10 million people annually. Currently, the pharmacotherapy used in this pathology works only in the acute phase, being ineffective in the chronic phase and eliminating the vector. Batroxycidin (BatxC), an α-helix peptide derived from the venom gland of the lancehead pit viper *Bothrops atrox*, was evaluated against the *T. cruzi* strain. BatxC inhibited the development of the benznidazole-resistant strains of the protozoan. In addition, it could lead to protozoan necrosis, observed by the loss of the cell membrane [71]. 

Among the parasitic diseases, Leishmaniasis is the seventh most important and represents a severe public health problem, affecting about 15 million people annually. This parasitosis is caused by the bite of the sand fly, mainly of the genera *Phlebotomus* and *Lutzomyia* [72]. Notably, histatin 5 (Hst5), an AMP isolated from human saliva that targets fungal mitochondria, showed anti-leishmania effects against strains of *Leishmania donovani*. In protozoa, Hst5 exerts its anti-parasite effects against promastigote and amastigote phases. This peptide proved capable of causing damage to the plasma membrane of the parasite, translocating to the *Leishmania* cytoplasm independently of a receptor for internalization. The peptide still has the potential to reduce ATP synthesis, leading to a subsequent drop in ATP content and causing a bioenergetic breakdown. Thus, Hst5, in addition to exhibiting anti-leishmanial effects, acts directly on the mitochondrial ATP synthesis of this protozoan [73]. Tachypesin is a peptide derived from the horseshoe crab *Tachypleus tridentatus* of marine origin, containing 17 residues in its structure with a positive charge and molecular weight of 2.36 kDa. When evaluated in the same strain mentioned above, this substance demonstrated efficacy in the promastigote and amastigote forms of *L. donovani*, damaging the membrane in a way that makes it difficult for the parasite to acquire resistance [74]. Phospholipases A2 (PLA2) are potent active toxins with multiple biological activities from enzymatic to neurotoxic, so their structure is used as a model for the structural design of several molecules with numerous pharmacological properties.

The PLA2-derived p-AclR7 peptide and analogs were synthesized and evaluated against strains of *Leishmania amazonensis*. The peptide p-AclR7 was designed from the C-terminal p-Acl region of a segment of Lys49 from the PLA2 from the venom of the broad-banded copperhead snake *Agkistrodon contortrix laticinctus*. An analog prepared by replacing all its lysine residues with arginine combined the characteristics of its original p-Acl and oligoarginine. The replacement of lysine for arginine residues in the peptide sequence potentiated the antiparasitic effect compared to the original peptide. The p-AclR7 was most active against the amastigote and promastigote stages of the protozoa [75]. Peptides with antimicrobial activities were evaluated in two other *Leishmania* species, namely *L. panamensis* and *L. major*. From the waxy monkey leaf frog *Phyllomedusa sauvagei*, it was possible to isolate dermaseptin, a potent AMP with a broad spectrum of action against fungi and bacteria. Dermaseptin promoted an increase in nitric oxide levels and acted on intracellular forms of the *Leishmania* parasite [76]. Cecropin A is a peptide isolated from various insects with sequences with 35–37 amino acids in length. Cecropin A and andropin prevented the development of intracellular forms of *Leishmania* in a dose-dependent fashion, demonstrating a direct activity of these peptides and an action related to the activation of some cellular functions of the parasitized phagocyte [77]. Derived from cecropin, CM11 is a hybrid peptide consisting of 11 residues, 4 of which are in the domain five of melittin (6 to 9 residues) C-terminal and cecropin A (2 to 8 residues) in the N-terminal region. The anti-leishmanicidal activity was observed in the *Leishmania* significant strain and exhibited a dose-dependent effect on promastigotes, in addition to demonstrating substantial effects on amastigotes [77]. As recently reviewed, the effectiveness of the members of the principal families of AMPs exhibiting a potential anti-leishmanial activity is discussed [78].

The examples listed in Table 3 give a glimpse of cell-penetrating peptides with anti-parasite activity and AMPs with cell-penetrability to kill intracellular disease-causing protozoa.

## 6. Discussion

The cumulative data in the current scientific literature place the CPAPs in a particular group of biological multivalent active peptides with a high potential of druggability. The cell-penetrating peptides with anti-infective activity and AMPs with cell penetrability behavior offer uncountable possibilities for pharmaceutical development to care for intracellular infection caused by difficult-to-treat pathogenic microorganisms. Whether on one side, microorganisms evolved strategies to evade the host immune by coopting the cellular machinery and hiding inside the cell and subcellular compartments, on another side, the drugs have limited efficacy to translocate across cellular membranes and act intracellularly, consequently contributing to increasing microbial drug-resistance [8,10]. The primary mechanism of action of AMPs is disrupting the cytoplasmic membrane of microorganisms and even cancer cells [79,80,81,82]. However, a handful of AMPs exert their effect intracellular by homing to organelles like the mitochondria and the nucleus and interacting with protein and nucleic acids [16,32,83].

In contrast, CPPs translocate across cytoplasmic and lipid membranes by different mechanisms, like endocytosis-dependent and energy-independent, without causing membrane damage [84,85]. In the translocation process, CPPs can transport hydrophilic substances and nanoparticles, serving as an intracellular shuttle and delivery system. By homing in the cytoplasm or other cytoplasmic compartments of target cells where, for instance, infectious intracellular microorganisms temporally reside or conditions in which the cell cycle is altered, as in cancer cells, CPPs can serve as molecular scalpels to interrupt cellular processes coopted by intracellular microorganisms.

Making evident the growing list of multivalent CPAPs, that is, cell-penetrating with anti-infective (antibacterial, anti-parasite, and antiviral) and anti-proliferative (anticancer) activities and antimicrobial peptides with cell-penetrating capabilities, the clinical arsenal to target-driven intracellular infection is substantially improved. CPPs covalently conjugated with antibiotics and antivirals also appear as an excellent pharmaceutical strategy to deal with intracellular infectious agents [17], while chimeric peptide sequences emerging as a rational design of hybrid CPAPs [78]

It is a matter of debate that AMPs and CPPs display low selectivity toward diseased and healthy cells, narrowing the therapeutic index. However, some critical level of selectivity does exist, notably, the composition and charge of membranes of bacteria, fungi, and parasite, the lipid envelope of the virus, and the membrane of cancer cells, which facilitate the interaction of bioactive peptides [80,86,87]. Interestingly, some CPAPs internalize preferentially infected and diseased cells, as inferred from the abovementioned examples. This fact is evidenced by the low level of cytotoxicity caused by CPAPs to healthy host cells in contrast with cells loaded with parasites in vitro models. As one can observe, this particular class of bioactive peptides, CPAPs, bears two essential functionalities in a single molecule: cell penetrability and antibiosis. These functionalities can be combined with other properties of the CPPs and the AMPs, like receptor-mediated endocytosis and conjugation [88,89] and interconnection of innate and adaptative immunity [90,91].

## 7. Conclusions

This review summarizes cell-penetrating antimicrobial peptides with anti-infective activity against intracellular microorganisms. It covers a list of compelling studies demonstrating the promising antimicrobial effects of peptides that internalize infected cells and kill intracellular pathogens. Although CPAPs are druggable and amenable to be translated from the bench to the clinical sets, strategies of structural design and peptide engineering of promising CPAPs are advisable to increase their selectivity and stability for in vivo use.

## 8. Materials and Methods

This systematic review followed the PRISMA (Preferred Reporting Items for Systematic Reviews and Meta-Analysis) guidelines [92]. The PubMed, ScienceDirect, and Google Scholar electronic databases were inquired for published articles on cell-penetrating peptides with antimicrobial activities against intracellular infections. The bibliographic search on the database was up to October 2022. The interrogation of databases was without the restriction of the date of publication. The publications were analyzed using a search string containing the terms: “Cell-penetrating peptides” in combinations such as “Intracellular infections”, “Antimicrobial peptide”, “Intracellular parasite”, “Antiviral”, “Malaria”, “Leishmania”, and “Trypanosoma”. The selection of works was carried out with the collaboration of the reviewers and through Mendeley software (version 1803, 2020) and verified, ensuring the review work’s quality. The selected literature followed the criteria: full research articles conducted in vitro or in vivo experimental studies and evaluated cell-penetrating peptides as an antimicrobial agent. In addition, other studies containing reports related to the topic of this review were selected that were found outside of the original search. The criteria used to exclude studies were repeated articles, reviews, editorials, letters to the editor, theses, dissertations, reports, and articles that do not agree with the subject of this review. The papers selected for inclusion in the systematic review were chosen by the authors who added studies that followed the required criteria. The information collected from the literature contains the following information: authors, year of publication, peptides used in the study, amino acid sequences, target microorganism, study model, and main results. Thirty-three original articles were initially selected from the databases search using the primary keywords. References were further complemented with essential information on the topics covered in this review.

## Figures and Tables

**Table 1 antibiotics-11-01772-t001:** CPAPs with anti-infective activity against intracellular bacteria.

Peptide	Source	Sequence	Target	Study Model	Ref.
TPk	Neutrophil granules	VRRFkWWWkFLRR	*S. aureus*	in vitro	[16]
PenShuf	Synthetic peptide	RWFKIQMQIRRWKNKK	*S. aureus* and *E. coli*	in vitro	[16]
L2 and L10	A synthetic peptide with a fragment of the pheromone (DILIIVGG) of *Streptococcus agalactiae*	DILIIVGGGSGKERKKRRR DILIIVGGKRRR	*Streptococcus agalactiae*	in vitro and in vivo	[42]
LDP	*Lachesana tarabaevi* (Spider)	KWRRKLKKLR	MRSA, *Bacillus subtilis, Escherichia coli, Salmonella typhimurium, Xanthomonas oryzae, Mycobacterium smegmatis*	in vitro	[46]
TAT (47–58)	TAT protein of HIV-1	YGRKKRRQRRRD	MRSA	in vitro	[47]
H2	Derived from Plectasin, isolated from *Pseudoplectania nigrella*	-	MRSA	in vitro and in vivo	[48]
Fl-PRPLPL-5	Synthetic peptide	-	*A. baumannii*, *E. coli* O157, MRSA e VRSA	in vitro	[49]
P14-5L, P14-5B and P14-5C	Synthetic peptide	-	*Salmonella* sp., *Shigella* sp. and *Listeria* sp.	in vitro in vivo	[50]
Fl-PLPRPR-4	Synthetic peptide	-	*Salmonella typhimurium* and *Brucella abortus*	in vitro	[51]
Pep-1	Synthetic peptide	KETWWETWWTEWSQPKKKRKV	*Chlamydia trachomatis*	in vitro	[52]
[I11A]tachyplesin I and [I11S]tachyplesin I	Analogs from Tachyplesin I, a horseshoe crab peptide.	KWCFRVCYRGACYRRCR KWCFRVCYRGSCYRRCR	*Escherichia coli*	in vitro	[53]

**Table 2 antibiotics-11-01772-t002:** Cell-penetrating antimicrobial peptides with antiviral activity.

Peptide	Source	Sequence	Target	Study Model	Ref.
MFK	Synthetic peptide	MFKLRAKIKVRLRAKIKL	HIV-1 Virus	in vitro	[55]
TAT (48–57)	TAT protein of HIV-1 (Strain SF2)	GRKKRRQRRR	HIV-1 Virus	in vitro	[56]
LK-3	Synthetic peptide	LKKLCKLLKKLCKLAG	HIV-1 Virus	in vitro	[57]
NYAD-1	Synthetic peptide	-	HIV-1 Virus	in vitro	[58]
Target of Egr1 (TOE1)	Nuclear protein in nucleoli and Cajal bodies	335-KRRRRRRREKRKR-347	HIV-1 Virus	in vitro	[59]
P16/16	Synthetic peptide	CSPQYTIIADAGDFYLHPSYYMLRKRRKR	HPV Virus	in vivo and in vitro	[60]
Hp1036 and Hp1239	Venom of scorpion *Heterometrus petersii*	ILGKIWEGIKSIF ILSYLWNGIKSIF	Herpes simplex virus type 1	in vitro	[61]
P4	Synthetic peptide	LDPAFR	Hepatitis B virus	in vitro	[62]
Deca- (Arg)_8_	Synthetic peptide	Decanoic acid (C10:0)-D(WRRRRRRRRG)-NH2	Hepatitis B Virus	in vitro	[63]

**Table 3 antibiotics-11-01772-t003:** Cell-penetrating antimicrobial peptides against intracellular parasites.

Peptide	Source	Sequence	Target	Study Model	References
Crotamine	*Crotalus durissus terrificus* (Snake)	YKQCHKKGGHCFPKEKICLPPSSDFGKMDCRWRWKCCKKGSG	*Plasmodium falciparum*	in vitro	[64]
Crotamine	*Crotalus durissus terrificus* (Snake)	YKQCHKKGGHCFPKEKICLPPSSDFGKMDCRWRWKCCKKGSG	*Plasmodium falciparum*	in vitro	[65]
rR8-JR21	Derived from the junctional region of *Plasmodium falciparum*	rRrRrRRR-KKKKKKKKKYYKYKKKEKEKK	*Plasmodium falciparum*	in vitro	[67]
TP10	Synthetic peptíde	AGYLLGKINLKALAALAKKIL	*Plasmodium falciparum* and *Trypanosoma brucei brucei*	in vitro	[69]
(L-cyclohexyl alanin-D-arginine)_3_	Synthetic peptíde		*Plasmodium falciparum*	in vitro	[70]
Batroxicidin (BatxC)	*Bothrops atrox* venom gland	KRFKKFFKKLKNSVKKRVKKFFRKPRVIGVTFPF	*Trypanosoma cruzi*	in vitro	[71]
Histatin 5	Human saliva	DSHAKRHHGYKRKFHEKHHSHRGY	*Leishmania donovani*	in vitro	[73]
Tachyplesin	Marine-sourced Japanese horseshoe crab (Tachypleus tridentatus)	KWCFRVCYRGICYRRCRGK	*Leishmania donovani*	in vitro	[74]
p-AclR7	Synthetic peptide	RRYRAYFRFRCRR	*Leishmania (L.) amazonensis*	in vitro	[75]
Andropin and Cecropin A	Hemolymph of the giant silkworm *Hyalophora cecropia*	VFIDILDKMENAIHKAAQAGIG KWKLFKKIEKVGQNIRDGIIKAGPAVAWVGQATQIAK	*Leishmania panamensis*	in vitro	[76]
CM11	Chimeric peptide	WKLFKKILKVL	*Leishmania major*	in vitro	[77]

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
