# Peer review of "Cell-Penetrating Antimicrobial Peptides with Anti-Infective Activity against Intracellular Pathogens"

_antibiotics, 2022, doi:10.3390/antibiotics11121772_

Round 1
Reviewer 1 Report
This is a useful paper that should be published after addressing the following points:
- There are several reviews presenting the topic from different angles. These reviews seem to list well-known peptides which makes the coverage of this topic in the literature repetitive. This review suggests a broader scope including engineered sequences. These are presented as synthetic peptides. It is not clear if these sequences are de novo, re-engineered or modified wild-type sequences.
- Discuss recent progress made toward artificial and re-engineered sequences based on AMPs and CPPs to penetrate cells and target intracellular bacteria or inhibit their growth; 10.1111/j.1462-5822.2010.01563.x; 0.1021/acsabm.2c00741; 10.1039/d1bm01235e; 10.1038/s41467-017-02475-3; 10.1128/AAC.00685-09
- The review has a subsection on CPPs which is followed by a section on cell penetrating peptides with anti-infective activities. This is confusing and raises questions: what makes CPPs anti-infective? What makes a sequence cell penetrating and anti-infective? A subsection on antimicrobial peptides and a discussion on what makes CPPs and AMPs different can help the reader.
- In the subsection peptides that act against intracellular pathogens a new term introduced CPAPs. Include a subsection or examples of peptides that are active against extracellular but not intracellular pathogens.
Author Response
Rebuttal Letter _ Manuscript ID: antibiotics-826796
Response to reviewers and actions taken
First, on behalf of the coauthors, I sincerely thank the referees for their consideration, comments, criticisms, and suggestions.
The point-by-point answers are in the following lines of this letter.
The modifications on the article are indicate in yellow color.
Comments and Suggestions for Authors from Reviewer 1:
This is a useful paper that should be published after addressing the following points:
On behalf of coauthors, I appreciate the anonymous reviewer's understanding regarding our contribution. We amended the manuscript whenever possible to accommodate the reviewer's suggestions and consideration.
- There are several reviews presenting the topic from different angles. These reviews seem to list well-known peptides which makes the coverage of this topic in the literature repetitive. This review suggests a broader scope including engineered sequences. These are presented as synthetic peptides. It is not clear if these sequences are de novo, reengineered or modified wild-type sequences.
Authors' response and actions taken:
As mentioned in our review's main text, these bioactive peptides contain multivalent peptide sequences (multi-effectors). This review gives a glimpse into the possibility of investigating and developing natural and synthetic (de novo designed and chimeric) peptides that penetrate cells and display antimicrobial effects. Examples (not completely extensive) from the current literature are listed and referenced for illustration.
- Discuss recent progress made toward artificial and re-engineered sequences based on AMPs and CPPs to penetrate cells and target intracellular bacteria or inhibit their growth; 10.1111/j.1462-5822.2010.01563.x; 0.1021/acsabm.2c00741; 10.1039/d1bm01235e; 10.1038/s41467-017-02475-3; 10.1128/AAC.00685-09
Authors' response and actions taken:
A paragraph concerning the idea of activating macrophages using defensins was included (in: examples of CPAPs’ mechanisms of action).
- The review has a subsection on CPPs which is followed by a section on cell penetrating peptides with anti-infective activities. This is confusing and raises questions: what makes CPPs anti-infective? What makes a sequence cell penetrating and anti-infective? A subsection on antimicrobial peptides and a discussion on what makes CPPs and AMPs different can help the reader.
Authors' response and actions taken:
These sections were revised trying to combine the CPP backgrounds. However, we decided to keep in separate to distinguish the basic differences between classes.
One paragraph calling the attention about distinction of exclusive activities of CPPs and AMPs was included at the end of the section to enlighten the particular class of CPAPs (cell-penetrating antimicrobial peptides), i.e., cell-penetrating peptides with anti-infective activity or AMPs with cell-penetrating activity.
- In the subsection peptides that act against intracellular pathogens a new term introduced CPAPs. Include a subsection or examples of peptides that are active against extracellular but not intracellular pathogens.
Authors' response and actions taken:
CPAPs was defined in the text as Cell-penetrating antimicrobial peptides, i.e., cell-penetrating peptides with antimicrobial activity and vice-versa, antimicrobial peptides with cell-penetrating capabilities. Other authors refer to this class of bioactive peptides as “Antimicrobial and Cell-Penetrating Peptides” (Antibiotics 2022, 11(11), 1636; https://doi.org/10.3390/antibiotics11111636), so we mention this in the text.
A paragraph was included to emphasize that most AMPs were characterized against extracellular microorganisms.

Reviewer 2 Report
Uploaded file

Author Response
Rebuttal Letter _ Manuscript ID: antibiotics-826796
Response to reviewers and actions taken
First, on behalf of the coauthors, I sincerely thank the referees for their consideration, comments, criticisms, and suggestions.
The point-by-point answers are in the following lines of this letter. The modifications to the article are indicated in yellow color.
Comments and Suggestions for Authors from Reviewer #2:
Dear Editor, Silva Cruz et. al. have presented a comprehensive review on the role of cell-penetrating peptides (CPP) against intracellular pathogens. Using various examples, the authors describe CPP's antibacterial, antiviral and antiparasitic activities. The authors conclude that CPP, combined with chemotherapeutics, could serve as a better and more effective alternative to conventional antibacterial therapy.
The authors have presented the study in a detailed and clear manner. I do, however have some comments that perhaps the authors could address.
On behalf of the coauthors, I appreciate the anonymous reviewer's understanding regarding our contribution. We amended the manuscript whenever possible to accommodate the reviewer's suggestions and consideration.
- Line 107: Perhaps the authors could divide the intracellular pathogens section into bacteria ( gram positive then gram negative), parasites , viruses rather than clubbing them together. They can also try and summarize in a table.
Authors' response and actions taken:
Thank the reviewer for this suggestion. However, we decided not to divide microbial classes because some peptides are effective as antimicrobials in more than one class.
- Line 134: For better clarity i would suggest that the authors include a new paragraph, since the authors here are addressing the problem and providing rationale for use of CPP.
Authors' response and actions taken:
The paragraph was amended to become the proposition clear.
- Line 156: Since this is the first CPP, the authors could describe the MOA of TAT
Authors' response and actions taken:
Because the MOAs are diverse, references were included that direct the readers to selected reviews regarding CPPs to illustrate the structures and mechanisms of actions not only of TAT but other CP peptides.
- Line 161: Secondary structure
Authors' response and actions taken:
Correction made. Thanks
- Line 198: Please clarify if bacteria internalize macrophages or macrophages internalize bacteria and thus being protected from the host immune response.
Authors' response and actions taken:
The paragraph was rephrased to clarify what happens in bacteria internalization events. Thanks
